# Local Dynamic Path Planning for an Ambulance Based on Driving Risk and Attraction Field

**Fang Zong, Meng Zeng, Yang Cao \* and Yixuan Liu**

College of Transportation, Jilin University, Changchun 130022, China; zongfang@jlu.edu.cn (F.Z.); zengmeng18@mails.jlu.edu.cn (M.Z.); yixuanl19@mails.jlu.edu.cn (Y.L.)
\* Correspondence: yang_cao@jlu.edu.cn

**Abstract:** Path planning is one of the most important aspects for ambulance driving. A local dynamic path planning method based on the potential field theory is presented in this paper. The potential field model includes two components—repulsive potential and attractive potential. Repulsive potential includes road potential, lane potential and obstacle potential. Considering the driving distinction between an ambulance and a regular vehicle, especially in congested traffic, an adaptive potential function for a lane line is constructed in association with traffic conditions. The attractive potential is constructed with target potential, lane-velocity potential and tailgating potential. The design of lane-velocity potential is to characterize the influence of velocity on other lanes so as to prevent unnecessary lane-changing behavior for the sake of time-efficiency. The results obtained from simulation demonstrate that the proposed method yields a good performance for ambulance driving in an urban area, which can provide support for designing an ambulance support system for the ambulance personnel and dispatcher.

**Keywords:** traffic flow; ambulance; local planning; potential field

## 1. Introduction

In today's world, fortunately, it is possible to cure many diseases and injuries with the development of medical technology. However, obtaining timely aid is crucial for the patient, especially in accidents. Under such circumstances, an ambulance reaching the event scene in a timely manner has a significant economic and social influence [1]. There has been news reporting the death of patients due to the untimely rescue of ambulances. Issues such as the untimely departure of the ambulance, being unfamiliar with the routes and delays caused by congested traffic are the main reasons resulting in this. For example, statistics indicate that the chance of survival decreases by 24% for every additional minute of delay in the treatment of cardiac arrest patients [2]. Hence, seeking an effective path, especially in congested traffic, for ambulances is one of the notably demanding research areas because of its vital and direct impact on human lives.

At this point, as a sub-field of transportation in health care, seeking an effective emergency medical services (EMS) management plan has been widely studied [3–5]. Since those plans have a direct effect on human lives, many researchers are dedicated to optimizing them by satisfying all the constraints at the best point, such as minimizing responsiveness time, maximizing availability and so on. Although these plans seem reasonable and sufficient, they are not adapted to the microcosmic level. Factors such as the complex road situation, traffic jams during rush hour, etc., will complicate ambulance driving in the city, thereby influencing travel time. Hence, in addition to pre-determining a global route, a specific path plan corresponding to these dynamics is also expected.

Speaking of specific planning, there are many path planning methods, including artificial potential field, random search and optimal control [6–9]. Among them, artificial potential field, which was first proposed by [10], has the advantage of low calculation cost [11]. By assigning repulsive force for obstacles and attractive force for the goal position

virtually, the gradient of potential field is formed. Then, a path can be generated along the steepest gradient of the potential field. For the past few decades, the artificial potential field (PF) has been widely utilized for vehicles, including connected and autonomous vehicles and traditional ones. The driving risk, including road boundaries, traffic regulations and obstacles, and the attraction from the target and preceding vehicle were considered for modeling PF [12–15]. However, the developed PF models may not be suitable for ambulances as their driving characteristics differ from regular vehicles. For example, the ambulance always follows its leading vehicle tightly due to fast mobility. Additionally, the ambulance can ignore traffic regulations in some cases. In these situations, the PF models in previous studies may be inconsistent with the driving reality of ambulances. Moreover, a vehicle, especially an ambulance, should try to avoid unnecessary lane-changing behaviors in order to improve time-efficiency. Hence, the motion states of vehicles in other lanes would affect the lane-changing decision, which is also lacking consideration in the existing studies.

To address the limitations in the existing studies, this paper proposes a PF-based local dynamic path planning method for an ambulance in an urban traffic environment. A repulsive PF model is built to describe the potential risk of the traffic entities, including the road boundaries, lane lines, and obstacles. Likewise, the gravity in the driving environment is realized by an attraction PF model. Then, a simulation test is conducted to evaluate the proposed method.

This paper is organized as follows. A literature survey is presented in Section 2. The PF model is established in Section 3. Then, a simulation is employed and the results are discussed in Section 4. Finally, this paper ends with a conclusion, including suggestions for future works.

## 2. Literature Review

### 2.1. Ambulance Planning

The majority of the studies on EMS mainly focus on macroscopic planning. The planning related to location [16,17] and route problems [18–22] is primarily considered in previous studies. The location problem has been widely studied over the last few decades. The study of [23] is one of the first papers that addressed the emergency vehicle location problem. Following this, a sequence of studies on ambulance location was conducted by researchers [24–30]. For example, Ref. [31] proposed a mathematical model and algorithm for ambulance location and relocation. Ref. [32] explored the reallocation of idle ambulances and the variation in rescue coverage, and proposed a location model for ambulances. Ref. [33] considered the uncertain travel time and conducted three models, namely, a robust covering, expected covering and expected p-robust covering model.

Meanwhile, a successive study focuses on the route problem. For example, Ref. [34] proposed a variable neighborhood search algorithm to schedule a route for ambulances. They further extended the research by combining allocation with dynamic programming so as to minimize the aiding time [35,36] allowed for an urgent situation where a lot of injuries/patients simultaneously require urgent medical care, and proposed a swarm-based approach for solving the ambulance routing problem. Ref. [37] considered the uncertainty and established a dynamic ambulance routing model. In order to find the shortest path to the hospital or patient, Ref. [38] proposed a solution based on the Dijkstra algorithm while considering current traffic conditions. In Ref. [39], the shortest path is sought by an approximate dynamic programming method. Considering the limitation of reaching a destination before the deadline, Ref. [40] employed a cardinality minimization approach to solve the stochastic shortest path problem. The authors further extended their research by employing the reinforcement learning method to improve the accuracy of finding the optimal shortest path so as to minimize the probability of delay occurrence [41]. The results yielded a feasible performance over other methods. In addition, in order to improve the computation efficiency of the stochastic routing problem, they also proposed a partial lagrange multiplier method [42]. The experimental results indicated that the proposed method can efficiently solve the routing problem.

Conclusively, there is a wealth of studies for ambulance planning, addressing its efficient achievement. However, to the best of our knowledge, most of these studies are implemented from a macroscopic perspective. After pre-determining a global route, during the driving process, a complex local traffic situation should be allowed for, and correspondingly, local dynamic path planning should be implemented. Such microscopic planning is still lacking.

*2.2. Potential Field for Path Planning*

Potential field has been widely used for robotics to avoid obstacles [9,43]. For example, Ref. [44] characterized the repulsive energy of obstacles as an adaptive potential function in a multi-robot system. In recent years, with the development of intelligent transport systems, many researchers applied potential field to the path planning of vehicles. For example, Ref. [45] considered the constraints of road structure, traffic regulations and obstacles in the driving environment and established a longitudinal planning method based on potential field. Similarly, Ref. [46] improved the potential field by considering both the constraint, i.e., road structure, traffic rules and obstacles, and the attraction, i.e., target position and tailgating. They involved vehicles' velocity for autonomous path planning in a microscopic field but ignored the influence of lane-velocity to lane-changing. Conclusively there are some drawbacks to the existing potential field methods. Firstly, only a scant number of driving influence factors and their repulsive effects are considered, and the attraction elements are mainly limited to the target. Secondly, most applications of these PF-based methods are limited to regular vehicles. There exists a clear distinction between ambulances and regular vehicles, particularly under congested traffic. For example, owing to urgency, when the space between two vehicles in two adjacent lanes is passable, the ambulance can drive through it, ignoring the constraint of the lane line under congested traffic. Hence, these functions in previous studies are difficult to adapt to ambulances.

Heeding these limitations, we propose a local dynamic path planning method for an ambulance. A comprehensive potential model is used to represent the driving risk and attraction due to various traffic factors. The repulsive factors include road boundaries, lane lines and obstacles. As mentioned above, considering the difference between regular and emergency vehicles, especially in congested traffic, an adaptive potential function for lane lines is constructed in association with traffic conditions. As for attraction potential, in addition to target and tailgating potential, a lane-velocity potential is proposed to characterize the influence of velocity in other lanes on lane-changing. Using a simulation of a congested-traffic scenario, we will demonstrate the performance to evaluate the proposed method.

## 3. Potential Field Modelling

In a real traffic scenario, ambulance motion is influenced by many factors. Firstly, the ambulance is heading to the desired position such as the accident scene or the hospital. Secondly, the ambulance should stay within the road and preferably drive in the lanes. Thirdly, the average velocity in each lane usually influences a vehicle's lane-changing behavior. Fourthly, the dynamic characteristics of the ambulance and other traffic entities play a significant role in maintaining a safe distance between each other. The velocity of the ambulance and other entities is a key factor to determine the safe distance. Thus, considering the influence mechanism of each factor, the total potential $P_{total}$ can be divided into two components, i.e., repulsive potential $P_r$ (road boundary potential $P_{rb}$, lane line potential $P_{ll}$ and obstacle potential $P_{obs}$) and attractive potential $P_a$ (target potential $P_{tar}$, lane-velocity potential $P_{lv}$ and tailgating potential $P_{tai}$). As depicted in Equation (1), each component will be addressed below, respectively.

$$P_{total} = P_r + P_a = P_{rb} + P_{ll} + P_{obs} + P_{tar} + P_{lv} + P_{tai} \tag{1}$$

### 3.1. Repulsive PF Model

There exists many repulsive entities during the driving process, such as road boundaries, traffic regulations, static obstacles and moving objects, etc. We divide these repulsive entities into three categories, i.e., road boundaries, traffic regulations and obstacles. The purpose of building a repulsive potential field is to characterize the collision risk between the ambulance and these repulsive entities. Considering the constraint of vehicle motion, as depicted in Equation (2), the repulsive potential field comprises (a) the road boundary potential field, (b) the lane line potential field, and (c) the obstacle potential field.

$$P_r = P_{rb} + P_{ll} + P_{obs} \tag{2}$$

- Road Boundary Potential Field

During the driving process, the ambulance needs to allow for the constraints of the road boundaries to avoid lane departure and collision. Generally speaking, a road boundary is a hard limit as it cannot be crossed inherently, even if there is an emergency situation. It is conceivable that the closer the ambulance is to the boundary, the stronger the repulsive force generated by the edge is. Therefore, the magnitude of the potential field generated by the $i$-th road boundary ($i \in \{1, 2\}$, $i =1$ denotes the left road boundary, $i = 2$ right road boundary) for the ambulance position $(x,y)$ is calculated by:

$$P_{rb,\,i}(x,y) = A_{road}d_{r,i}^{s}{}^{-3} \tag{3}$$

where $P_{rb,\,i}(x,y)$ represents the PF value of the road boundary $i$ at the position $(x, y)$. $A_{road}$ is the coefficient of the road boundary potential field. $d_{r,i}^{s}$ is the shortest lateral distance from point $(x,y)$ to road boundary $i$ (m). $d_{r,i}^{s} = |y - y_{r,i}|$, $y - y_{r,i}$ denotes the lateral distance vector from the ambulance to the $i$-th road boundary, $y_{r,i}$ stands for the lateral position of the $i$-th road boundary, $i = 1, 2$. It is known that the magnitude of the field value is mainly determined by the lateral distance, and the repulsive energy is highest at each side of the road boundary.

- Lane Line Potential Field

Normally, in a structured road, in addition to the road boundary, the vehicles' motion largely depends on the constraint of lane lines, enabling them to drive along the center line. However, compared to the road boundary, the lane line is a soft constraint as it can be overcome in accordance with traffic regulations. Notably, for a regular vehicle, the constraint of lane lines works except for lane-changing. In contrast, for the ambulance, the repulsion of lane lines disappears, especially in the case of intense traffic. The reason is that in most incidents, the achievement of that means that an ambulance is able to reach either the event scene or hospital in a timely manner, which is the primary goal. Hence, when the traffic is at a low-density situation, the emergency vehicle can follow traffic rules to drive along the center line while ensuring efficiency. Once the traffic is congested, one of the rules is that the ambulance can ignore the constraint of lane lines and search for a "new lane" (the surrounding possible passable gap) to move forward. Thus, in this paper, we set a trigger for lane line potential associated with the threshold effect of traffic density.

$$P_{ll,\,i}(x,y) = INT[\delta \cdot sgn(0.28 - \mu) + 1]A_{lane}[-exp(y - y_{l,i}) + 1] \tag{4}$$

where $P_{ll,\,i}(x,y)$ stands for the PF value of the lane line $i$ at the position $(x,y)$. $A_{lane}[-exp(y - y_{l,i}) + 1]$ is the lane line potential function. $A_{lane}$ is the coefficient of the lane line potential. $y_{l,i}$ is the lateral position of the $i$-th lane line (m). $y - y_{l,i}$ denotes the lateral distance vector from the ambulance to the $i$-th lane line, which mainly determines the PF value. Notably, as mentioned above, the ambulance can violate the traffic rule and drive in the reverse lane in some congested situations. In such cases, the lane line potential is zero in this paper. Thus, $INT[\delta \cdot sgn(0.28 - \mu) + 1]$ is the activation trigger we propose for lane line potential associated with the traffic condition. $\mu$ is the traffic

condition indicator, referring to our previous study [47], $\mu > 0.28$ denotes that the traffic is congested, otherwise uncongested. $\delta$ is a constant between 0 and 1 that ensures the function of $INT[\delta \cdot sgn(0.28 - \mu) + 1]$ can only accept 0 or 1, and here it is 0.5. Then, when $\mu > 0.28$ (the traffic is congested), the function of $sgn(\cdot) = -1$, which means the function of $INT[\delta \cdot sgn(0.28 - \mu) + 1 = INT(0.5) = 0$. Then, the lane line potential is zero. Conversely, when $\mu \leq 0.28$ (the traffic is uncongested), the function of $sgn(\cdot) = 1 \ or \ 0$, which means the function of $INT[\delta \cdot sgn(0.28 - \mu) + 1 = 1$. Then, the lane line potential works, which matches the situation mentioned above.

A composition of road boundary potential and lane line potential is illustrated in Figure 1. It should be noted that running off the road is discouraged, so the barriers generated by the road boundary are the highest, then the double yellow line, and finally the lane line.

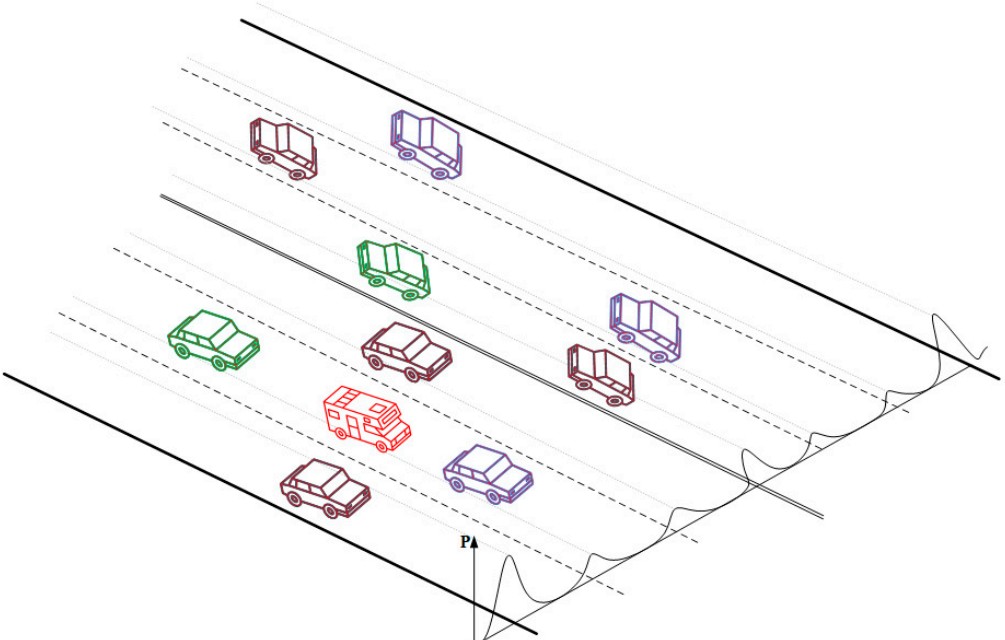

**Figure 1.** Illustration of road and lane barriers.

- Obstacle Potential Field

The potential field of obstacles derives from the entities that may affect the driving of the subject vehicle. The contribution of building the obstacle potential field is to depict the potential collision risk of the surrounding obstacle around the ambulance, which is essential for it to keep a safe distance from the obstacles. The magnitude of danger associated with obstacles depends on several factors, including the pure distance and speed. Pure distance refers to the spatial distance between the ambulance and the obstacles; the smaller the gap is, the greater the "repelling force" is. However, the lateral and longitudinal distance impose a different influence on the potential. It should be noted that given vehicles' dynamic and driving rules, the collision risk along the longitudinal distance is generally higher than that in the lateral direction as the obstacles, especially dynamic obstacles, will generate a speed vector opposite to the driving direction of the ambulance. Another factor affecting the strength of the repulsive potential is velocity, including absolute velocity and relative velocity. As the relative velocity increases, a corresponding longer safe distance is required. Similarly, the faster the ambulance, the stronger the repulsion it perceives. Then,

we formulate the obstacle potential field by Equation (5) [46]. The diagram of an obstacle potential field is shown in Figure 2.

$$P_{obs,j}(x,y) = A_{obs} \cdot exp\left\{-\left[\frac{(x-x_{obs,j})^2}{B_x\sigma_x^2} + \frac{(y-y_{obs,j})^2}{B_y\sigma_y^2}\right] + \xi\left(v - v_{obs,j}\right)\frac{(x-x_{obs,j})^2}{B_x\sigma_x^2}\right\}$$
$$\xi = \begin{cases} 0, x > x_{obs,j} \\ 1, x < x_{obs,j} \end{cases}$$
(5)

where $P_{obs,j}(x,y)$ represents the PF value of obstacle $j$ at the position $(x,y)$. $A_{obs}$ is the parameter of the obstacle potential field. $(x_{obs,j}, y_{obs,j})$ is the position of the $j$-th obstacle. $v$ is the velocity of the ambulance (km/h). $v_{obs,j}$ denotes the velocity of obstacle $j$ (km/h). $v_{obs,j} = 0$ when the obstacle is a static entity. As mentioned above, the higher the relative velocity is, the closer the distance from the ambulance to the obstacle is, and the stronger the generated repulsive energy is. Therefore, we use the relative distance both in longitudinal $(x - x_{obs,j})$ and lateral direction $(y - y_{obs,j})$ and the relative velocity $v - v_{obs,j}$ to characterize the potential field. $\sigma_x$ and $\sigma_y$, respectively, stand for the influence range of an obstacle in the longitudinal and lateral direction. Differing from the ordinary vehicle, the ambulance always tends to follow its leading vehicle tightly. Consequently, the influence range of an obstacle perceived by an ambulance is usually smaller than that a regular vehicle gains. Thus, we introduce a tolerance indicator ($B_x$ and $B_y$, which is, respectively, a constant between 0 and 1) to modify the influence zone [48]. $\xi$ is a regulation coefficient for changing the repulsive potential in the rear part of an obstacle vehicle; as all drivers sit facing the front, a leading driver usually does not respond to a situation happening behind, such as an approaching obstacle vehicle.

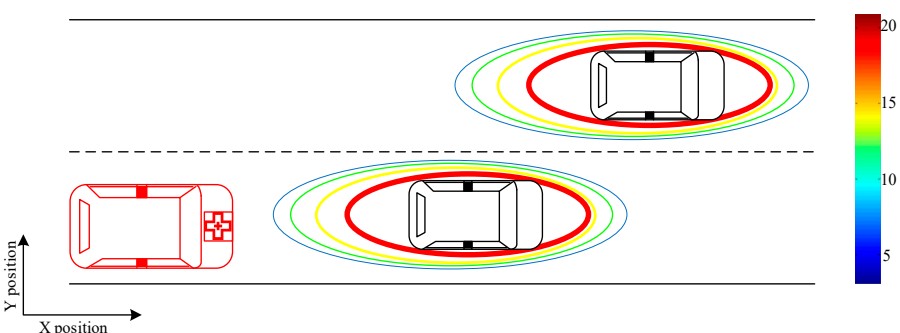

**Figure 2.** The diagram of obstacle potential field.

### 3.2. Attractive PF Model

Similarly, "attraction" exists in traffic flow. For example, when the velocity of the adjoining lane is faster, a vehicle will feel a "force" to change lane when driving with a relatively slow speed. The aim of constructing the attractive potential field is to describe the gravity in the traffic environment. Considering the vehicle dynamics and driving preference, in this paper, the attractive potential field refers to the composition of (a) the target potential field, (b) lane velocity potential, and (c) the tailgating potential field (shown in Equation (6)).

$$P_a = P_{tar} + P_{lv} + P_{tai}$$
(6)

- Target Potential Field

In real traffic scenarios, a vehicle will flow in a pre-determined direction, which is similar to the phenomenon of a free object falling to the ground. The reason is that it is constantly subject to earth gravity. Hence, it is reasonable to imagine that the ambulance is subject to a "gravity" from the event scene or hospital. Such force and consequent field are

related to the factor of distance. As shown in Equation (7), the level of attraction increases as the distance between the ambulance and the target increases [46].

$$P_{tar} = -A_{target}exp(x - x_{tar}) \qquad (7)$$

where $P_{tar}$ is the PF value of the target at the position $(x,y)$. $A_{target}$ is the coefficient of target potential. $x_{tar}$ is the position of the target in longitudinal direction (m). $x - x_{tar}$ is the distance from the ambulance to the target in $x$ direction, which mainly determines the magnitude of the PF value.

- Lane Velocity Potential

As mentioned before, an ambulance's first strategy of moving on the road is to achieve mobility. Hence, it is recognized that the velocity in a lane will generate potential; the faster the velocity in another lane is, the greater the lane-changing attraction is to the ambulance. Therefore, the velocity potential of the *i*-th lane acting on the ambulance in the longitudinal direction can be expressed as:

$$P_{lv,\, i} = -A_{lv}\tau \frac{\overline{v}_{lv,i} - \overline{v}}{\overline{v}} \qquad (8)$$

$$\tau = \begin{cases} 0, \overline{v} > \overline{v}_{lv,i} \\ 1, \overline{v} < \overline{v}_{lv,i} \end{cases} \qquad (9)$$

where $P_{lv,\, i}$ stands for the PF value of velocity in lane $i$. $A_{lv}$ stands for the coefficient of lane-velocity potential. $\overline{v}_{lv,i}$ is the average speed in the *i*-th lane. $\overline{v}$ is the average speed in the current lane (km/h). Generally speaking, in order to improve time-efficiency, a vehicle should avoid unnecessary lane-changing behaviors as much as possible. However, in real driving scenarios, instead of observing the whole motion state of all the vehicles in a lane, a vehicle is easily affected by the surrounding vehicles' motion and conducts unessential lane-changing, reducing driving efficiency. Thus, in this paper, we consider the motion state of other lanes to provide support for lane-changing decisions. The average velocity in a lane is chosen to represent the motion situation of all the vehicles in it. The greater the average velocity difference between two lanes is, the greater the lane-changing attraction generated by the lane-velocity is. $\tau$ is an indicator that ensures the lane-velocity will disappear when the average velocity of other lanes is smaller than that in its current lane.

- Tailgating Potential

During the driving process, an ambulance always tailgates (i.e., following at a relatively short distance), and it is likely that a temporal attraction point behind the preceding vehicle $m$ is perceived by the ambulance, which may respond by speeding up to follow it tightly. By incorporating a "tailgating attraction point", the tailgating potential in the $(x)$ direction can be determined as:

$$P_{tai,m}(x,y) = -A_{tai}exp[\frac{(x - x_{tai,m})^2}{B_x\sigma_x{}^2} + \frac{(y - y_{tai,m})^2}{B_y\sigma_y{}^2}] \qquad (10)$$

$$\begin{cases} x_{tai,\, m} = x_m - vt \\ y_{tai,m} = y_m \end{cases} \qquad (11)$$

where $P_{tai,m}(x,y)$ represents the PF value of a proceeding vehicle $m$ at position $(x,y)$. $(x_{tai,m},\, y_{tai,m})$ is the tailgating attraction position, which is calculated by Equation (11). $(x_m,\, y_m)$ is the coordinate of the preceding vehicle $m$ in the current lane. $t$ usually takes 3s [46].

In general, the integral object for the ambulance is to achieve mobility and safety (gains) while avoiding collision and the violation of traffic regulations (losses). Hence, such an object can be represented using an overall potential field $P_{total}$, which consists of repulsive field and attraction field in Equation (1).

If $P_{total}$ is interpreted as a mountain whose elevation denotes the risk, the ambulance's strategy is to navigate the mountain range along its valley, i.e., the least risky route.

## 4. Simulation Construction

### 4.1. Simulation and Parameter Design

Combing the above, a successive effort is required to test the validity of the proposed model. Then, as depicted in Figure 3, a driving scenario of an urban road with two lanes is constructed in Matlab. In the simulation, for simplicity, the lanes are set to be straight and we input many vehicles until the traffic density is high, with a local congestion index exceeding 0.28. The initial states of each vehicle, including velocities (30 km/h), position, and so on, are inputted. Then, a path for the ambulance will be produced by using the method in each time step (5 s). The parameter values used in the potential field are employed from [46], and the coefficient of lane-velocity potential was calibrated by the NGSIM (Next Generation Simulation) data. We calculated the lane-changing frequency under different velocities to obtain the value of the lane-velocity potential coefficient. The parameters are listed in Table 1.

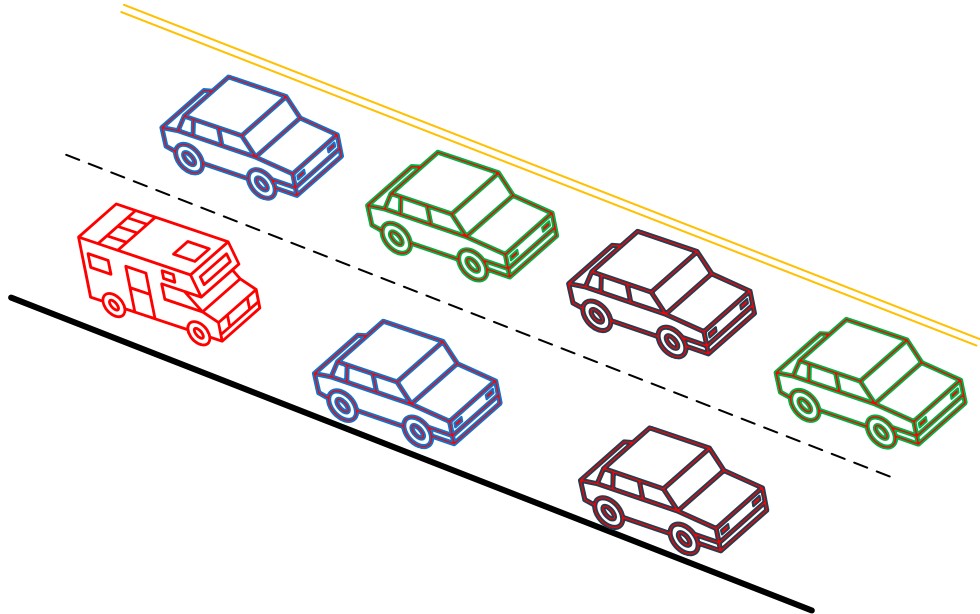

**Figure 3.** Basic simulation scenario.

**Table 1.** Parameter lists.

| Potential | Object | Parameter | Definition | Value |
|---|---|---|---|---|
| Repulsive potential | Road boundary | $A_{road}$ | Coefficient of road boundary potential | 20.00 |
| | Lane line | $A_{lane}$ | Coefficient of lane line potential | 1.00 |
| | Obstacle | $A_{obs}$ | Coefficient of obstacle potential | 10.00 |
| | | $\sigma_x$ | Convergence coefficient in $x$ direction | 10.00 |
| | | $\sigma_y$ | Convergence coefficient in $y$ direction | 1.50 |
| Attractive potential | Target | $A_{target}$ | Coefficient of target potential | 0.25 |
| | Lane velocity | $A_{lv}$ | Coefficient of lane-velocity potential | 0.14 |
| | Tailgating | $A_{tai}$ | Coefficient of tailgating potential | 10.00 |

*4.2. Results Discussion*

4.2.1. Simulation Result

In the simulation scenario, when the traffic density is high with a local congestion index exceeding 0.28, there is no opportunity for lane-changing. In such situations, the ambulance will seek to cut in and drive along center line with the vanishing of the lane line constraint. Figure 4 illustrates the desired trajectory calculated by the proposed method.

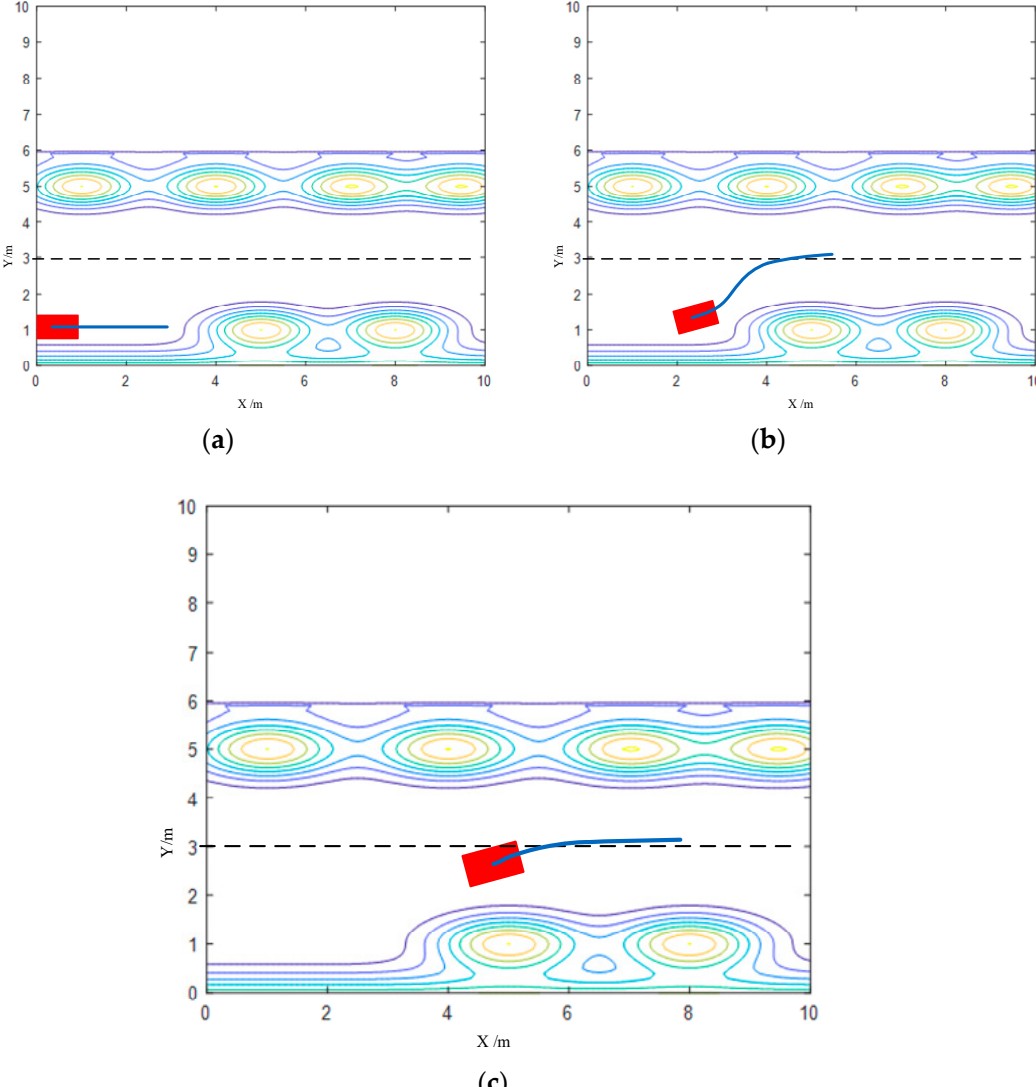

**Figure 4.** These three figures depict the driving process: (**a**) The ambulance is running in the right lane; (**b**) The ambulance starts to change to a "new lane"; (**c**) The ambulance drives along "new lane" successfully.

Figure 4a–c depicts the cut-in and collision avoiding process. Lastly, the ambulance evades risk and keeps running successfully. In the beginning, the ambulance keeps running in its current lane (Figure 4a). Owing to the congestion, the desired behavior of the ambulance is to seek a "new lane" generated by the disappearance of lane line potential. As shown in Figure 4b, the ambulance captures a "new lane" and the desired trajectory is generated to start moving forward to the lane line. Figure 4c shows that the ambulance cuts in successfully and keeps running along the lane line.

The trajectory generated by the proposed method is visualized in Figure 5, where the blue dash line denotes the trajectory that is driven by the ambulance, and the blue solid line is the current desired trajectory. Conclusively, the trajectory generated by the proposed

method is collision-free and realizes the mobility efficiency as much as possible under the congested condition.

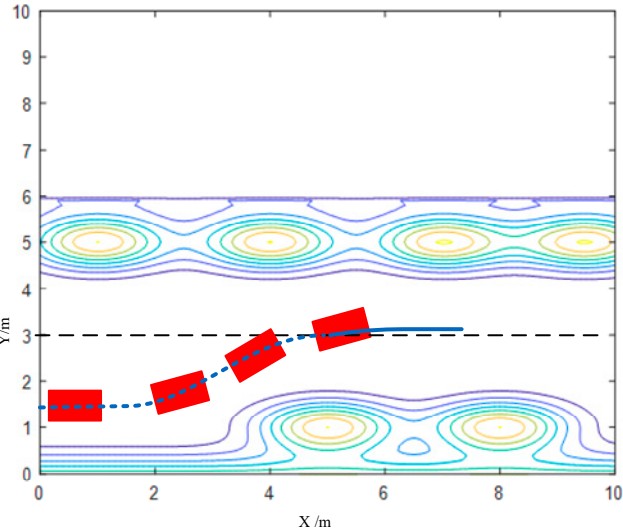

**Figure 5.** Generated trajectory.

### 4.2.2. Results Comparison

In order to verify the feasibility of our model, we conducted 15 simulations. Considering the objects in this method, we also compared the results to that without the proposed method. Figure 6 illustrates the comparison results with regard to path-length efficiency, i.e., path length with vs. without the method and time efficiency, i.e., travel time with vs. without the method. Table 2 demonstrates the basic statistic results.

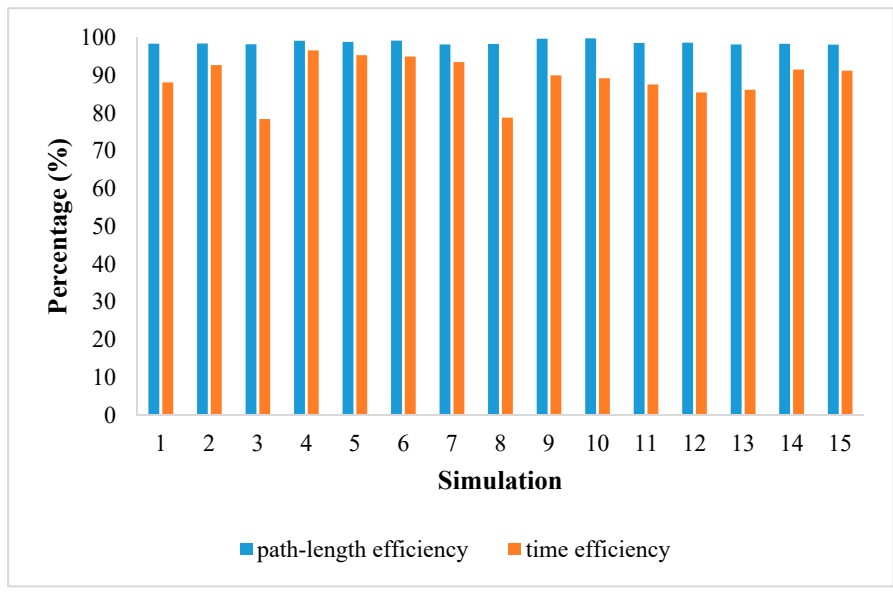

**Figure 6.** Results comparison.

As we see in Figure 6, the path length in our method is shorter than that without the method in each simulation, consequently resulting in a shorter travel time. Specifically, the average path length is shortened by 1.32%, and the average travel time is reduced by 10.68%, remarkably. The underlying reason is that when the gap between two adjacent vehicles is available for passing a vehicle, the ambulance can drive along the "new lane" by our method so as to shorten travel time drastically.

**Table 2.** Basic statistics.

| Item | Max | Min | Variance | Mean |
|---|---|---|---|---|
| Path-length efficiency (%) | 99.81 | 98.15 | 0.31 | 98.67 |
| Time-efficiency (%) | 96.61 | 78.42 | 29.97 | 89.32 |

## 5. Conclusions

In this paper, we propose a PF-based local path planning method for an ambulance in an urban traffic environment. Firstly, the potential risk of the traffic entities is described by a repulsive potential field, which comprises three parts: road boundary potential, lane line potential and obstacle potential. In addition, an attractive potential field is built to characterize the "gravity" in the traffic environment, which takes the target gravity, lane-velocity and tailgating attraction into consideration. With the combined potential field, a simulation is constructed to verify the performance of the proposed method. The simulation results show that the presented PF model yields a feasible performance.

The repulsive potential in this paper is built to describe the possible risk of the traffic environment, including road boundaries, traffic rules and obstacles. Notably, in a real driving scenario, we realize there is a driving distinction between ambulances and regular vehicles. That is, the ambulance can ignore the constraint of lane lines, especially in congested traffic. Hence, an adaptive function of lane line potential is constructed in association with the traffic situation. When the traffic is at a low-density situation, the emergency vehicle can follow traffic rules to avoid lane departure as the lane line potential exists. Once the traffic is congested, the ambulance is allowed to ignore the constraint of lane lines and search for the possible gap so as to continue moving forward. The simulation results indicate that the trajectory generated by the presented method is collision-free and realizes the mobility efficiency as much as possible under the congested condition.

In the specific context of attractive potential, different from the existing studies on path planning based on field potential, in addition to the target gravity and tailgating attraction, the influence of velocity in other lanes is also considered in the mesoscale planning to prevent unnecessary lane-changing behavior for the sake of time-efficiency. The results in Figure 6 indicate that the proposed method improves time-efficiency.

Notably, there are still some problems to be solved in the future. At first, an actual vehicle test in a real traffic environment should be conducted in the future. Moreover, if there is an ambulance, other vehicles usually make way for it. So, the collaboration of other vehicles should be allowed in future research. In addition, for simplicity, the lanes in the simulation are set to be straight, thus, more complex road environments should be involved in future simulations.

**Author Contributions:** Conceptualization, F.Z.; conceptualization, methodology, M.Z.; software, Y.L.; validation, formal analysis, F.Z. and Y.C.; writing—original draft preparation, M.Z.; writing—review and editing, F.Z.; supervision, Y.C. All authors have read and agreed to the published version of the manuscript.

**Funding:** This research was funded by the National Key R&D Program of China, grant number 2018YFB1600500 and National Natural Science Foundation of China, grant number 61873109.

**Conflicts of Interest:** The authors declare no conflict of interest.

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
