# Peer review of "Local Dynamic Path Planning for an Ambulance Based on Driving Risk and Attraction Field"

_sustainability, doi:10.3390/su13063194_

Round 1
Reviewer 1 Report
The study proposed a local dynamic path planning for ambulance to avoid accident risk and improve travel speed. Therefore, the ambulance could deliver emergency service for the customers. The planning considers the repulsive factors and attractive factors at the same time. The topic is very interesting and helpful to advance future autonomous ambulance vehicles, which are different from regular autonomous vehicle operations. The study can be improved by revising the abstract and providing details of the methodology.
The abstract can be updated with findings and benefits of the study to communities. Who are the communities to take advantage of the study. That would help readers to capture the study quickly from the abstract. The repulsive factors were introduced in Introduction but the attractive factors are not addressed. Those are always bundled together in Literature Review and Methodology. So consistency can be maintained. The repulsive and attractive factors in Section 3 should be listed and explained consistently. The equations and notations are understandable, but they are too briefly described for readers.
Section 4 modeling constructing should be improved in terms of model execution and design. The authors calibrated the model in order to find the parameters, but it didn’t explain how the calibration was done and how the model was benchmarked or tested. The parameters were explained with equations, but the units were not clears (e.g. Table 1). Do the parameters have distribution? How did you determined the number of runs of 15? If the model runs multiple times, the output should have variation and average on Figure 6. In addition, the limitation of the study is not fully described. The model assumed straight lanes of the roads for the purpose of the simulation. What are the computational speed while vehicle moving fast? What are the size of trajectory? 5 seconds or 10 seconds?
There are several minor changes as follows:
Line 269: Scenario 1? So are there other scenarios?
Line 143: (l) can be added for clarification after the sentence
Figure 1: The distributions for each lane and boundary were visualized at the end of right hand side. In the middle, the median should be the highest barrier, but the size of distribution is lower than the side lines. Any reason?
Line 124 and 125: the notations can be switched for readability.
Table 1: the columns of Potential and Object are not well aligned. Separate lines would be helpful.
Author Response
Please see the attatchment

Reviewer 2 Report
The authors exploit potential field theory to study the local path planning for ambulance. My concerns are as follows,
1- Please highlight the novelty of this work. Simply adapting the potential field method to the ambulance path planning may not be adequately qualified as a publication;
2- It seems that the authors only compare with themselves. They claim that rare works have been conducted for ambulance local path planning. If so, (1) whether this topic is of significance? or whether it is significantly different from the regular ones? (2) the authors may compare with one of the methods in [39-43], or the ones proposed for autonomous vehicles (I believe there are many similar ones for local path planning).
3- Regarding the path planning for ambulance, one more key criterion is Arriving on Time, which has been well addressed in 'Using Reinforcement Learning to Minimize the Probability of Delay Occurrence in Transportation. IEEE Transactions on Vehicular Technology' 'Finding the Shortest Path in Stochastic Vehicle Routing: A Cardinality Minimization Approach. IEEE Transactions on Intelligent Transportation Systems' 'Improving the Efficiency of Stochastic Vehicle Routing: A Partial Lagrange Multiplier Method. IEEE Transactions on Vehicular Technology'. They aim to maximize the probability of reaching the destination before the deadline, which suits well for ambulance path planning, the authors may include them.
4- the writing needs to be further polished. some issues as follows,
a. 'distinguish' in abstract seems to be a verb;
b. 'key core' seems redundant;
c. 'we use filed theory' should be' field'.
d. ...
Author Response
Please see teh attatchment
